# Adapting a guided low-intensity behavioural activation intervention for people with dementia and depression in the Swedish healthcare context (INVOLVERA): a study protocol using codesign and participatory action research

Frida Svedin ,[1] Anders Brantnell ,[1,2] Paul Farrand ,[3] Oscar Blomberg ,[1] Chelsea Coumoundouros ,[1] Louise von Essen ,[1] Anna Cristina Åberg ,[4,5] Joanne Woodford [1]

For numbered affiliations see end of article.

**Correspondence to**
Dr Joanne Woodford;
joanne.woodford@kbh.uu.se

## ABSTRACT

**Introduction** Dementia is a worldwide health concern with incident rates continuing to increase. While depression prevalence is high in people with dementia and psychological interventions such as cognitive behavioural therapy (CBT) are effective, access to psychological interventions remains limited. Reliance on traditional CBT for people with dementia and depression may present difficulties given it is a complex psychological approach, costly to deliver, and professional training time is lengthy. An alternative approach is behavioural activation (BA), a simpler psychological intervention for depression. The present study seeks to work with people with dementia, informal caregivers, community stakeholders, and healthcare professionals, to adapt a guided low-intensity BA intervention for people with dementia and depression, while maximising implementation potential within the Swedish healthcare context.

**Methods and analysis** A mixed methods study using codesign, principles from participatory action research (PAR) and normalisation process theory to facilitate the cultural relevance, appropriateness and implementation potential of the intervention. The study will consist of four iterative PAR phases, using focus groups with healthcare professionals and community stakeholders, and semi-structured interviews with people with dementia and informal caregivers. A content analysis approach will be adopted to analyse the transcribed focus groups and semi-structured interviews recordings.

**Ethics and dissemination** The study will be conducted in accordance with the Declaration of Helsinki and data handled according to General Data Protection Regulation. Written informed consent will be obtained from all study participants. In accordance with the Swedish Health and Medical Services Act, capacity to consent will be examined by a member of the research team. Ethical approval has been obtained from the Swedish Ethical Review Authority (Dnr: 2020-05542 and Dnr: 2021-00925). Findings will

## Strengths and limitations of this study

► To the best of our knowledge, this is the first study in Sweden integrating codesign, principles from participatory action research (PAR), and normalisation process theory (NPT) to develop a psychological intervention for depression in people with dementia.

► Including healthcare professionals and community stakeholders can help ensure the intervention is truly clinic and patient centered due to their 'real-world' knowledge and ability to provide useful information concerning how to efficiently implement the intervention.

► Utilisation of implementation theory, that is, NPT, should improve implementation potential and potentially optimise effectiveness.

► People with dementia do not need to have past or present depression to participate, hence we may recruit people with dementia who have no experience of depression or low mood, which may impact on transferability or results.

► With limited resources, this study will not be able to include ethnic minority groups due to the increasing costs, for example, recruitment, informed consent, and need for translated materials.

be published in an open access peer-reviewed journal, presented at academic conferences, and disseminated among lay and healthcare professional audiences.

## INTRODUCTION
### Background
Dementia is a worldwide health concern, with the number of people living with dementia set to rise from approximately 44 million

BMJ

in 2016[1] to over 140 million by 2050.[2] In Sweden, over 140 000 people are living with dementia,[1] with around 25 000 incident cases each year.[3] Further, the societal cost of dementia in Sweden is high and was estimated at nearly 63 billion SEK during 2012.[3] Due to both prevalence rates and the significant burden placed on individuals and wider society, dementia represents a challenge for global healthcare policy,[4] especially given the lack of curative or preventative medical interventions.[5] Indeed, dementia is expected to represent the largest increase in serious health-related suffering and burden globally between 2016 and 2060,[6] and is associated with increased mortality,[7] poor quality of life,[8] and poor functional outcomes.[9] Further contributing to health-related suffering are elevated symptoms of depression, with prevalence rates estimated to be between 30% and 87%.[10–13] Depression in people with dementia is associated with increased mortality,[14] poor quality of life,[10] sleep difficulties,[15] and reduction in activities of daily living.[10] Despite depression being highly prevalent and evidence-based psychological interventions such as cognitive behavioural therapy (CBT) being effective for people with mild-to-moderate dementia,[16] access to evidence-based psychological interventions remains limited.[17] Lack of access is of particular importance since pharmacological approaches for depression show poor efficacy and negative side effects in people with dementia.[18 19]

However, this psychological 'treatment gap' is not unique to the dementia population, with only 30% of people across Europe with mental health difficulties accessing support.[20] To address the psychological treatment gap, there have been global efforts to increase access to psychological interventions through the provision of low-intensity CBT (LI-CBT).[21 22] LI-CBT can be characterised as a single-strand approach, adopting a single CBT technique to target specific common mental health difficulties where an evidence base has been demonstrated.[23] Further, LI-CBT is delivered in a self-help format via a health technology (eg, written workbooks, audio books, internet-administered, smartphone applications),[21] with the provision of face to face, telephone, or e-mail guidance associated with larger effect sizes.[23] For example, across England, guidance to LI-CBT is provided by a psychological practitioner workforce[23] with graduate or postgraduate level training in the competencies required to support LI-CBT.[24]

LI-CBT may represent a solution to improve access to support for people with dementia and depression. One evidence-based LI-CBT technique for depression is simple behavioural activation (BA). The simple BA model seeks to target behavioural avoidance (eg, disengagement from pleasurable, routine, and necessary activities),[25] a common symptom of both depression[26] and dementia,[27] by increasing engagement in activities. As such, BA may also help overcome sedentary behaviours common in older adults,[28] which may lead to improved functioning in people with dementia (eg, engagement in, and ability to perform, activities of daily life). Further, the simple

BA model is a straightforward psychological approach[29] and may be easier for people with dementia to understand than more complex CBT techniques. In addition to showing promise for people with dementia and depression, BA may also benefit informal caregivers[30] given improved functioning in the person with dementia is associated with decreases in caregiver burden.[30] Informal caregivers may also engage more in activities with the person with dementia, possibly improving caregivers' own mental health.[31] This is of particular importance given global and social care priorities to support people with dementia to remain at home for as long as possible[32] have resulted in an increased reliance on support from informal caregivers.[33]

Given the promise of BA, a research programme, informed by phase I (development) of the Medical Research Council (MRC) complex interventions framework,[34 35] has been undertaken in England.[29] However, before applying the intervention in Sweden, there is a need to adapt the intervention to fit into the Swedish cultural[36] and healthcare context,[37] while maintaining fidelity to the intervention.[38] Cultural adaptation of an intervention refers to the systematic modification of the intervention considering language, culture, and context to ensure its compatibility with the target population's cultural patterns, meanings, and values.[39] An example of a language difference is the meaning of the term 'depressed', which in Swedish is associated with a formal diagnosis of depression, whereas in English the term is used more colloquially.[40] Consideration of context can include a number of characteristics that could impact intervention delivery and effectiveness and may include geographical, sociocultural, socioeconomic, ethical, legal, and political factors.[41 42] One contextual difference between England and Sweden is at present there is no psychological practitioner workforce in Sweden specifically trained in supporting LI-CBT.[43] As such, an appropriate healthcare professional workforce will need to be identified to facilitate intervention delivery. In addition, in England, national non-profit organisations, eg, Alzheimer's societies, provide support such as cognitive rehabilitation, information for informal caregivers, and activity groups, whereas in Sweden, support is predominantly provided via formal health and social care services rather than non-governmental organisations.[44] However, to the best of our knowledge, little research has examined potential cultural, language, and contextual differences between England and Sweden in the context of psychological intervention development and adaptation. Consequently, it is difficult to anticipate in advance what cultural adaptations may be required.

In addition, it is increasingly recognised that there are multiple facilitators and barriers to the implementation of evidence-based interventions, for example, problems with adaptation, fidelity, and delivery capability.[45] Normalisation process theory (NPT) provides a framework for understanding key mechanisms that facilitate or hinder the implementation of a complex intervention.[46–48] NPT

proposes four key constructs that enable the normalisation of a complex intervention by stakeholders: (1) coherence (individual and collective sense-making); (2) cognitive participation (engagement to build and sustain practice); (3) collective action (enact practices); and (4) reflective monitoring (appraisal).[47 49] Traditionally, NPT has been used to analyse the implementation of complex interventions.[50] However, more recently NPT has been used to inform the development of complex interventions to enhance implementation potential.[51 52] In the present study, we will use NPT throughout the adaptation process to gain an understanding of what may facilitate or hinder the implementation of the intervention and identify implementable evidence-based methods of intervention delivery.

## Aim and objectives

The overall aim is to adapt the guided low-intensity BA intervention developed in England for people with dementia and depression and their informal caregivers and enhance implementation potential for the Swedish cultural and healthcare context. Objectives are to:

1. Develop an understanding of the existing healthcare and community support context for people with dementia and their informal caregivers.
2. Identify barriers and facilitators to intervention uptake.
3. Identify feasible, acceptable and implementable evidence-based methods of intervention delivery informed by NPT.
4. Adapt the guided low-intensity BA intervention to ensure cultural appropriateness, relevancy, and acceptability to people with dementia and their informal caregivers.

## METHODS AND ANALYSIS
### Study design

A mixed methods study using codesign and principles from participatory action research (PAR),[53 54] placing key stakeholders (eg, people with dementia, informal caregivers, healthcare professionals, and community stakeholders) at the centre of the research process.[53] The study commenced on 23 March 2021 and the planned end date is 23 March 2022.

### Setting

Sweden is divided into 290 municipalities and 21 regions. Regions are predominantly responsible for general healthcare,[55] with municipalities responsible for general and specialised care and services in the home, nursing homes, day care, respite care, and support for informal caregivers.[56] This study will seek to involve key stakeholders from both general and specialised health and social care services. People with dementia and informal caregivers will be recruited via four specialised health and social care service settings across the county of Uppsala: primary healthcare centres; memory clinic; day care centres; and via dementia care consultants. Uppsala county incudes

both rural and urban areas and approximately 390 000 residents[57] with more than 5 000 people estimated having a diagnosis of dementia.[58] Healthcare professionals and community stakeholders from non-profit organisations will be recruited from locations across Sweden.

## Study participants
### Healthcare professionals and community stakeholders

Healthcare professionals (n≈8), eg, primary care and hospital-based physicians involved in dementia diagnosis, dementia care consultants, memory clinic nurses, nursing assistants working with people with dementia in day care centre settings, and clinical psychologists, will be recruited. Community stakeholders (n≈8) from relevant non-profit organisations will also be recruited, for example, from Dementia League, Alzheimer Sweden, and Red Cross.

### People with dementia and informal caregivers

The following inclusion criteria will be applied for people with dementia (n≈10):

1. Have a self-reported diagnosis of dementia (any type). The research team will not confirm the diagnosis by checking medical records.
2. Live at home.
3. Able to and have the capacity to provide informed consent, indicating mild to moderate dementia.[59]
4. Able to speak and understand Swedish.
   People with dementia will be excluded if they have:
1. A self-reported diagnosis of a severe and enduring mental health difficulty (eg, psychosis, type I or II bipolar disorder, and personality disorder).
2. A visual or auditory impairment that would hinder their ability to participate in semi-structured interviews and/or give feedback on the intervention.
3. A self-reported misuse of alcohol or prescription or street drugs reported by the potential participant as so severe that it interferes with their ability to perform normal activities in daily life.

The following inclusion criteria will be applied for informal caregivers (n≈10):

1. Self-identified informal caregiver of a person with dementia with whom they have regular contact (at least weekly).
2. Eighteen years of age or older.
3. Able to speak, understand, and write in Swedish.
   Informal caregivers will be excluded if they have:
1. A self-reported diagnosis of a severe and enduring mental health difficulty (eg, psychosis, type I or II bipolar disorder, and personality disorder).
2. A visual or auditory impairment that would hinder their ability to participate in semi-structured interviews and/or give feedback on the intervention.
3. A self-reported misuse of alcohol or prescription or street drugs reported by the potential participant as so severe that it interferes with their ability to perform normal activities in daily life.

## Sample size considerations

Sample size will be guided by thematic data saturation and as such we cannot stipulate the exact sample size in advance.[60 61] Interviews and focus groups will be analysed iteratively and a decision concerning whether thematic saturation is met will be made during analysis. However, we anticipate to include n≈8 healthcare professionals, n≈8 community stakeholders in each focus group,[62] n≈10 people with dementia and n≈10 informal caregivers, respectively, to participate in semi-structured interviews.

## Recruitment

### Healthcare professionals and community stakeholders

Healthcare professionals and community stakeholders will be identified via in person networks, social media, posters on notice boards, and printed flyers.[63] A member of the research team will speak to interested stakeholders over the telephone to provide more information alongside hand out a study invitation pack by post or e-mail containing: (1) a study invitation letter; (2) a study information sheet; (3) a reply slip; (4) reasons for non-participation questionnaire; and (5) a stamped addressed envelope.

### People with dementia and informal caregivers

A multifaceted approach to recruitment will be adopted across the following four settings:

#### Primary healthcare centres

Primary healthcare centres who are part of the network of academic primary healthcare centres will appoint a research nurse to conduct patient record searches to identify potentially suitable people with dementia. When identified, a research nurse will make contact and ask verbal permission, face to face or via telephone, for a member of the research team to contact them. If permission is granted, a member of the research team will contact the person with dementia over the telephone to provide more study information and send a study invitation pack (as above) by post to interested potential participants. The study invitation pack will also include an envelope containing a separate study invitation pack to be passed on to an informal caregiver.

#### Memory clinic

A physician will provide people with dementia and/or informal caregivers with brief verbal study information alongside hand out a study invitation pack (as above) during meetings whereby a person receives a diagnosis of dementia or follow-up visits (≤12 months since diagnosis).

#### Day care centres

Day care centre staff will provide informal caregivers with brief verbal study information and hand out a study invitation pack (as above).

#### Dementia care consultants

Dementia care consultants will provide people with dementia and/or informal caregivers with brief verbal study information and hand out a study invitation pack (as above).

## Informed consent and eligibility screening

### Healthcare professionals and community stakeholders

Interested healthcare professionals and community stakeholders respond to the study invitation by returning a reply slip, telephoning, or e-mailing the research team. A member of the research team will speak to those expressing interest in participation. During this phone call, interested stakeholders will have the opportunity to ask further questions about the study. Written informed consent will be obtained either face to face or via the post or alternatively, verbal recorded consent will be obtained over the telephone or via secure videoconferencing system. After provision of informed consent, background and sociodemographic characteristics—age, gender, profession, length of time in profession, professional qualifications, and length of time working with dementia—will be collected.

### People with dementia and informal caregivers

Interested people with dementia and informal caregivers can respond to the research team by returning a reply slip, telephoning, or e-mailing the research team. Those expressing interest in participation will be contacted by a member of the research team who will arrange a meeting (either face to face, over the telephone, or via secure videoconferencing system) to gain informed consent and undertake an eligibility screen. In accordance with the Swedish Health and Medical Services Act,[64] capacity to consent in people with dementia will be assessed by a member of the research team, with appropriate knowledge and training, at the meeting. People with dementia will be provided with appropriate support to maximise ability to provide informed consent,[65] for example, the provision of dementia-friendly written information.[66] Written informed consent will be obtained for people with dementia and informal caregivers met face to face and verbal recorded informed consent will be obtained by those met over telephone or secure videoconferencing.

After provision of informed consent, an eligibility screening will take place with questions asked in accordance with the inclusion/exclusion criteria. Given people with dementia may find it difficult to provide these data, an informal caregiver will be asked to attend the meeting and provide proxy data if needed.

## Reasons for non-participation

Study invitation packs will include anonymous reasons for non-participation form for those who decline participation. The reasons for non-participation form consists of a short questionnaire including possible reasons for non-participation informed by previous research[67 68] and an open-ended question. Reasons for non-participation will provide important information regarding recruitment feasibility and the acceptability of the proposed intervention.[68 69]

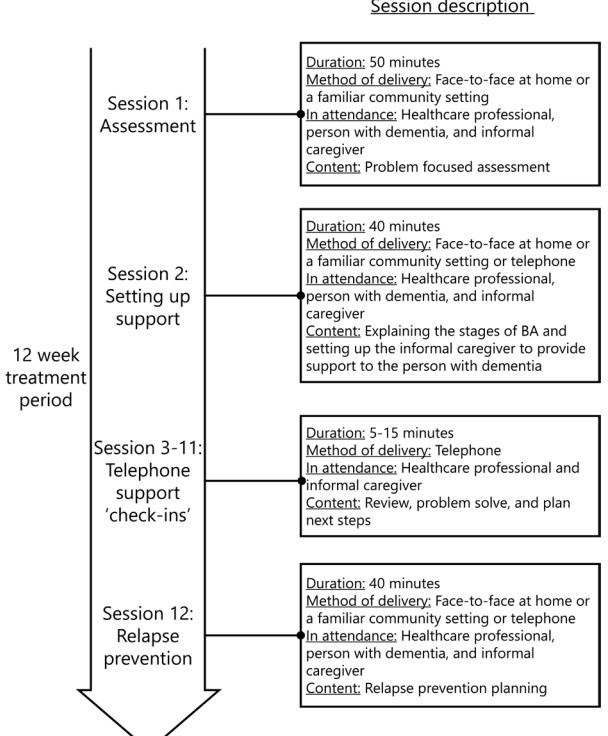

Session description

**Figure 1** Overall model of delivery of the low-intensity behavioural activation (BA) intervention developed in England.

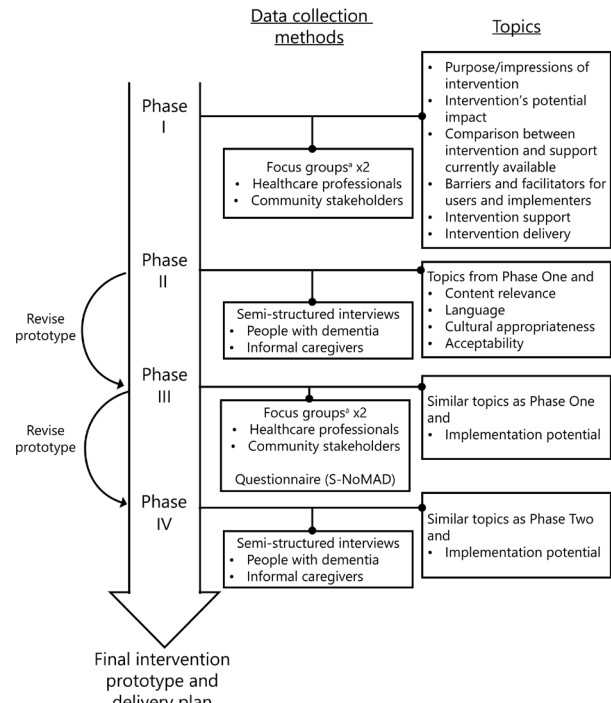

Data collection methods        Topics

**Figure 2** An overview of the data collection procedure and topics for the four participatory action research phases. S-NoMAD, Swedish version of the Normalisation Process Theory Measure.
**a Semi-structured interview if not able to attend focus group.**

## Intervention

The full clinical protocol for the low-intensity BA intervention developed for people with dementia in England can be found elsewhere.[29] Therapeutic content follows a simple BA intervention protocol.[21] Activities are categorised into three types: (1) routine (activities providing life structure, eg, housework); (2) pleasurable (activities providing a sense of pleasure or enjoyment determined by the person with dementia, eg meeting with friends); and (3) necessary (activities that if not done have the potential for serious negative consequences, eg taking medication or paying a bill). People with dementia are supported to either re-engage with activities they have stopped doing or replace activities they are no longer able to do (eg due to symptoms of dementia) with activities of similar value, importance, and meaning. Re-engaging in activity is done in a structured and gradual manner, with an overall aim to establish a balance of activities. The simple BA protocol includes four main steps: (1) identifying current activities; (2) identifying stopped activities or new activities; (3) organising activities; and (4) planning activities. Two workbooks are provided to help support intervention delivery. One workbook[70] is designed for the person with dementia to work through the steps of BA. The second workbook[71] is designed for an informal caregiver to help support the person with dementia work through the steps of BA. The overall model of delivery is shown in figure 1. The intervention is delivered using a maximum of 12 sessions over a period of 3 months. The intervention is guided by healthcare professionals who have been trained in the competencies needed to support LI-CBT self-help[21 72] and an informal caregiver also provides support to the person with dementia to work through the intervention. Session one (assessment) and session two (setting up support) are attended by the healthcare professional, person with dementia, and informal caregiver. Thereafter, the informal caregiver receives weekly telephone support 'check-ins' with the healthcare professional to review difficulties, intervention progress, problem solve, and plan the next steps. The final session (relapse prevention) is attended by the healthcre professional, person with dementia, and informal caregiver.

## PAR phases

Four PAR phases will be conducted, with estimated time frame of 1 month for each phase. An overview of the data collection procedure and topics for the four PAR phases can be found in figure 2.

### PAR phase I

Two focus groups (90–120 min) will be held with healthcare professionals and community stakeholders, with semi-structured interviews offered to those unable to attend a focus group. Two members from the research team will organise and facilitate the focus groups, one as a moderator and one as an observer. Prior to the focus group/interview, participants will be provided with a written summary of the model of delivery developed in

England and translated copies of the two workbooks in Swedish.

A topic guide, informed by NPT,[47 49] will consist of open-ended questions examining:

1. Perceived purpose of the intervention.
2. First impressions of the intervention.
3. Potential impact of the intervention on people with dementia and informal caregivers.
4. Perceived differences and similarities to existing support available for people with dementia and informal caregivers.
5. Barriers and facilitators to people with dementia and informal caregivers using the intervention (eg, travel and mobility, physical health problems, stigma, informal caregiver resistance, COVID-19 restrictions).
6. Potential professional/non-professional group to support and guide people with dementia and informal caregivers using the intervention (eg, dementia care consultants, nurses, psychologists, volunteers).
7. Preferred intervention delivery setting (eg, home, primary healthcare centre, memory clinic, day care).
8. Type of support and guidance (eg, telephone, face to face, online) needed to facilitate people with dementia and informal caregivers using the intervention.
9. Anticipated barriers and facilitators to the potential professional/non-professional group implementing the intervention (eg, resources, training, managerial support, change agents).

Focus groups/interviews may be held face to face, via videoconference, or over the telephone, dependent on preference and will be audio recorded. The research team will keep a reflective journal, logging the progresses, obstacles, and successes of the research process.[73]

### PAR phase II

Individual semi-structured interviews will be held with people with dementia and informal caregivers, respectively. Prior to the interview, participants will be provided with a written summary of the model of delivery developed in England and translated copies of the two workbooks in Swedish. A topic guide, informed by NPT,[47 49] will examine similar topics to those examined in PAR phase I. Additional focus will be placed on the perceived relevance of the intervention content and language and ways to enhance relevancy, cultural appropriateness, and acceptability. Interviews (45–60 min) will be held face to face (in a convenient location for the person with dementia and informal caregiver, eg, in their home or other familiar community location chosen by the participant), or via videoconference/telephone.

### PAR phase III

Based on PAR phases I and II results, a new version of the intervention and proposed model of delivery (eg, professional/non-professional group to support and guide people with dementia and informal caregivers using the intervention, intervention delivery, and type of support and guidance) will be developed. Two focus groups will be held with healthcare professionals and community stakeholders from PAR phase I, with semi-structured interviews offered to those unable to attend. Two members from the research team will organise and facilitate the focus groups, one as a moderator and one as an observer. A topic guide, similar to the one used in PAR phase I, will guide focus groups/interviews and examine the implementation potential of the new version of the intervention in accordance with NPT. At the end of the focus groups/interviews, the Swedish version of the NPT Measure (S-NoMAD)[74] will be completed by all healthcare professionals to examine implementation potential. The S-NoMAD[74] consists of 20 items corresponding to the four NPT constructs (coherence, cognitive participation, collective action, and reflexive monitoring)[47] in relation to the potential implementation of the proposed BA intervention. Responses are provided using a five-point Likert Scale, ranging from 'strongly disagree' to 'strongly agree'. 'Neutral' and 'not applicable' are also given as options.[74] The internal consistency of the S-NoMAD (Cronbach's alpha 0.76–0.83) are in line with the original NoMAD.[75]

### PAR phase IV

A second round of semi-structured interviews with the same people with dementia and informal caregivers who participated in PAR phase II will be conducted. A new version of the intervention and proposed delivery model established in PAR phases I–III will be presented. A topic guide, similar to the one used in PAR phase II, will guide interviews and examine the implementation potential of the new version of the intervention in accordance with NPT.

### Data analysis

Focus groups/interviews will be transcribed and uploaded into NVivo V.12.0 to support data analysis. Data analysis will be iterative, eg, conducted after each PAR phase to inform the subsequent phase. A content analysis approach will be adopted to analyse transcribed recordings.[76] NPT constructs will inform deductive coding, however, inductive coding will be applied to relevant data unrelated to NPT constructs. The S-NoMAD results will be analysed descriptively[74] in SPSS V.25. Triangulation by data source, where data sources are assessed against one another to crosscheck data and aid interpretation,[77] will be conducted by exploring agreements and disagreements across the qualitative (focus groups/interviews) and quantitative (S-NoMAD) data. Comparisons between the qualitative and quantitative data sets will be discussed in the research team to explore potential reasons for discordant data.

### Trustworthiness

Trustworthiness will be established via the use of a trustworthiness checklist developed for content analysis.[78] To ensure study credibility, peer examination (eg, researchers discussing the research process and findings with impartial and experienced colleagues), field

journals, and member checking (eg, continuously testing with informants themes, interpretations and conclusions) will be conducted. Dense description and presenting demographic data will be incorporated to ensure transferability.[77] Two members of the research team will independently identify codes and themes[77] and thereafter compare their findings. Codes and themes will also be discussed by the wider research team and patient and public involvement (PPI) group (see section Patient and public involvement).

### Reporting

Qualitative results will be reported through the Standards for Reporting Qualitative Research (SRQR).[79] Intervention content will be reported in accordance with the Template for Intervention Description and Replication (TIDieR).[80] PPI will follow Guidance for Reporting Involvement of Patient and Public 2 (GRIPP2).[81]

### Patient and public involvement

A PPI group consisting of people with dementia (n=3) and informal caregivers (n=3) will be established to work alongside the research team as active research partners throughout the research process. The PPI group will be recruited via advertisement through existing networks, relevant online/social media groups, and relevant local healthcare settings. The PPI group will meet together with the research team once every 2–3 months, with meetings held face to face, via video conference, and/or telephone depending on group members' preferences. The PPI group will work alongside the research team with research activities, such as advising on recruitment and research procedures, codevelopment of the intervention materials, interpretation of study results, and dissemination. A member of the research team will log all PPI activities in an impact log, which will be distributed to PPI group members immediately after each meeting to comment on accuracy.[82]

### DISCUSSION

At present, the psychological support needs of people with dementia and depression living at home are not met by the Swedish healthcare system.[83] To the best of our knowledge, this will be the first study to develop and adapt a low-intensity BA intervention for people with dementia in the Swedish setting. The proposed intervention has the potential to meet unmet psychological needs and also global priorities to support people with dementia to 'live well' with dementia and promote 'healthy ageing'. The intervention has further potential to reduce informal caregiver burden by providing effective strategies for informal caregivers to support people with dementia and depression.[31] The planned study is designed to improve the implementation potential of the intervention by using NPT. Given barriers to the implementation of evidence-based healthcare interventions are common, NPT will provide a framework for understanding potential barriers

and facilitators to implementation from the intervention development phase. This will enhance future implementation potential should the intervention be demonstrated to be clinically and cost effective in the future. Our study design also allows us to collect in-depth qualitative data to provide a rich understanding of how to adapt the intervention from the perspective of people with dementia and informal caregivers. Our careful exploration of the perceived relevance of the intervention content and language and ways to enhance relevancy, cultural appropriateness, and acceptability will facilitate the development of a more acceptable and relevant intervention, specifically tailored to needs and preferences of the population.

Despite the aforementioned study strengths, there are some limitations. First, people with dementia do not need to have past or present depression to participate. As such, we may recruit people with dementia who have no experience of depression or low mood, thus limiting the transferability of findings to people with dementia and depression. However, we will ask people with dementia about their past and present well-being, providing us with some indication as to the experience of depression in the study sample. Second, due to resource limitations, we are unable to include ethnic minority groups who do not speak Swedish due to the increasing costs, eg, recruitment, informed consent, and need for translated materials. This will limit the transferability of results and sample representativeness, for example, for people with dementia and a migration background who are at increased risk of marginalisation and unfortunately there is currently a lack of culturally appropriate interventions for this population.[84] However, the innovative intervention development process adopted in this study has potential for wider applications and can serve as a template for future adaptations of the intervention for people with dementia and other populations not represented in the present study.

In conclusion, this study will result in the development of a tailored intervention, hopefully optimised to improve relevancy and acceptability for a currently neglected population. Results will be used to inform the design of a phase II (feasibility) following the MRC framework for the development and evaluation of complex interventions[35] to further explore the feasibility and acceptability of the intervention.

### ETHICS AND DISSEMINATION

The study will be conducted in accordance with Declaration of Helsinki.[85] All data will be handled according to General Data Protection Regulation.[86] Informed consent will be obtained from all study participants[87] and capacity to consent in people with dementia will be established in accordance with Swedish Health and Medical Services Act.[64] Capacity to consent will be examined by a member from the research team. Given only people with dementia with capacity to consent will be included in this study,

we anticipate people with mild-to-moderate dementia will participate. The study is approved by the Swedish Ethical Review Authority (Dnr: 2020-05542 and Dnr: 2021-00925). Findings will be published in an open access peer-reviewed journal. We will aim to present results at scientific conferences as well as publications for public and professional audiences. The research team and PPI group will also promote results via social media.

**Author affiliations**
[1]Healthcare Sciences and e-Health, Department of Women's and Children's Health, Uppsala University, Uppsala, Sweden
[2]Division of Industrial Engineering and Management, Department of Civil and Industrial Engineering, Uppsala University, Uppsala, Sweden
[3]Clinical Education Development and Research (CEDAR), University of Exeter, Exeter, UK
[4]Geriatrics, Department of Public Health and Caring Sciences, Uppsala University, Uppsala, Sweden
[5]Medical Science, School of Education, Health and Social Studies, Dalarna University, Falun, Sweden

**Contributors** FS: methodology, resources, writing—original draft—and project administration. AB: writing—original draft, project administration and supervision. PF: conceptualisation and writing—review and editing. ACÅ: methodology and writing—review and editing. OB: writing—review and editing—and project administration. CC and LvE: writing—review and editing. JW: conceptualisation, methodology, writing—original draft—supervision, project administration and funding acquisition.

**Funding** This work was funded by the Swedish Research Council (Dnr: 2018-02691). CC was funded by the European Union's Horizon 2020 Research and Innovation Programme under the Marie-Skłodowska Curie grant agreement no. 814072.

**Disclaimer** Funders were not involved in the creation, development or publication of this protocol. Funders will not be involved in the conduct, analysis or reporting of the resulting study.

**Competing interests** None declared.

**Patient and public involvement** Patients and/or the public were involved in the design, or conduct, or reporting, or dissemination plans of this research. Refer to the Methods section for further details.

**Patient consent for publication** Not required.

**Provenance and peer review** Not commissioned; externally peer reviewed.

**ORCID iDs**
Frida Svedin http://orcid.org/0000-0002-8421-4908
Anders Brantnell http://orcid.org/0000-0001-6841-7644
Paul Farrand http://orcid.org/0000-0001-7898-5362
Oscar Blomberg http://orcid.org/0000-0002-7472-5130
Chelsea Coumoundouros http://orcid.org/0000-0001-5539-974X
Louise von Essen http://orcid.org/0000-0001-5816-7231
Anna Cristina Åberg http://orcid.org/0000-0001-8196-0553
Joanne Woodford http://orcid.org/0000-0001-5062-6798

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
