## [Reviewer comments · BMJ Open]

ARTICLE DETAILS

TITLE (PROVISIONAL)	Adapting a guided low-intensity behavioural activation intervention for people with dementia and depression in the Swedish healthcare context (INVOLVERA): a study protocol utilising co-design and participatory action research
AUTHORS	Svedin, Frida; Brantnell, Anders; Farrand, Paul; Blomberg, Oscar; Coumoundouros, Chelsea; von Essen, Louise; Åberg, Anna; Woodford, Joanne

VERSION 1 – REVIEW

REVIEWER	Xu, Xin Yi
REVIEW RETURNED	25-Feb-2021

GENERAL COMMENTS	Thank you for inviting me to review this nice paper. This study has been explained in a detailed and comprehensive way. The application of PAR and NTP gives a solid foundation for the development of study protocol. However, it would be better to include the proposed BA in this study protocol, so that readers could have a better understanding of the BA intervention. In addition, the discussion is a bit short. A more detailed discussion should be included. I'm looking forward to seeing your further study.
--

REVIEWER	Subramaniam, Ponnusamy
REVIEW RETURNED	26-Feb-2021

GENERAL COMMENTS	-Overall, this is well written protocol Introduction; last paragraph; - Good for reader to know, if the author can provide more information on the anticipated cultural barrier in adapting BA intervention from UK to Sweden. Any previous work(s) reported cultural barriers in adapting psychological intervention from UK to Sweden. Apart from language, what are the other cultural barriers. Any possible difference in knowledge, attitudes and practices (KAP) between people with dementia (also caregiver, other stakeholders) in UK and Sweden which may led to current proposed adaptation. This will be very interesting for readers, especially from outside Europe. Methods & Analysis
---

-Not sure how researcher come out with required participant for their study. Healthcare professionals (n=8), community stakeholders (n=6), PwD (n=10), caregiver (n=10) etc. Any sample size calculation was carried out for focus group or rule of thumb approach? Please provide detail or justification how researchers determine those numbers.

-Good to state the four settings at 'Setting' first.

- Inclusion and exclusion criteria;

-Probable diagnosis? Not confirmed diagnosis? Please clarify because this protocol aims BA for depression in PwD.

-Only person with mild to moderate dementia or this study will include those with severe dementia. It is challenging to implement intervention for those people with severe dementia. If only mild to moderate dementia, how the researcher determines the severity e.g. CDR? MMSE?

-Any screening to identify the present of depression in PwD. How about level of depression? Although depression is common in PwD but not all PwD have depression. Any criteria? Because this proposal is BA for PwD and with Depression.

-How about frailty level in PwD. Those people with dementia but robust & pre-frail may able to participate in BA program. Do physical condition of PwD matter for this protocol?

- Page 11, line 9 to 11 and line 22 to 24 are repetition on invitation/information pack. Please revise to avoid repetition.

-Page 11, line 17-18; ' A research nurse....against the inclusion...' So, in medical record the research will screening suitable PwD for study? Should the researcher team screen against the inc/exc criteria after identifying study participant (PwD), right? Early in page 10 (Inc/exc criteria), stated e.g. 'self-reported misuse...' This only can establish after interview or talk with caregiver and PwD. Please improve the clarity.

-Page 13, line 7, please check the sentence's wording.

-Good if can provided a table with flow/data collection with timeframe/timing (days or weeks or months) for by Phase (1, 2, 3 & 4). Good to have or indicate study participants involved for each phase. Having this summary table will useful for reader.

-Data analysis;

-What is s-NoMAD? Please give brief detail about that tool for reader e.g. items, scoring etc?

-S-NoMAD is to triangulate qualitative findings? Not sure how, please provide brief information to reader.

Thank you!

VERSION 1 – AUTHOR RESPONSE

Reviewer 1

Thank-you for your very helpful comments. Please find how we have addressed each comment below.

- 1 The application of PAR and NTP gives a solid foundation for the development of study protocol. However, it would be better to include the proposed BA in this study protocol, so that readers could have a better understanding of the BA intervention.***

We have now added a sub-section “intervention” within the Methods and Analysis section and provided further detail about the BA intervention developed in England, which will form the basis for this intervention adaptation study (see page 12, lines 13-33 and page 13, lines 1-8):

Intervention

The full clinical protocol for the Low-Intensity BA intervention developed for people with dementia in England can be found elsewhere.[29] Therapeutic content follows a simple BA intervention protocol.[21] Activities are categorised into three types: (1) routine (activities providing life structure, e.g., housework); (2) pleasurable (activities providing a sense of pleasure or enjoyment determined by the person with dementia); and (3) necessary (activities that if not done, have the potential for a serious negative consequences e.g., taking medication, or paying a bill). People with dementia are supported to either re-engage with activities they have stopped doing, or replace activities they are no longer able to do (e.g., due to symptoms of dementia) with activities of similar value, importance, and meaning. Re-engaging in activity is done in a structured and gradual manner, with an overall aim to establish a balance of activities. The simple BA protocol includes four main steps (identifying current activities, identifying stopped activities or new activities, organising activities, and planning activities). Two workbooks are provided to help support intervention delivery. One workbook[70] is designed for the person with dementia to work through the steps of BA. The second workbook[71] is designed for an informal caregiver to help support the person with dementia work through the steps of BA. The overall model of delivery is shown in Figure 1. The intervention is delivered using a maximum of 12 sessions over a period of 3 months. The intervention is guided by healthcare professionals who have been trained in the competencies needed to support CBT self-help[21, 72] and an informal caregiver also provides support to the person with dementia to work through the intervention. Session One (Assessment) and Session Two (Setting Up Support) are attended by the healthcare professional, person with dementia, and informal caregiver. Thereafter, the informal caregiver receives weekly telephone support ‘check-ins’ with the healthcare professional to review difficulties, intervention progress, and problem solve, and plan the next steps. The final session (Relapse Prevention) is attended by the healthcare professional, person with dementia, and informal caregiver.

Figure 1. Overall model of delivery of the Low-Intensity BA intervention developed in England.

2 In addition, the discussion is a bit short. A more detailed discussion should be included.

We have now added some further detail to the discussion, including an appreciation of the strengths and limitations of the study design (see page 17, lines 3-33 and page 18, lines 1-10):

At present, the psychological support needs of people with dementia and depression living at home are not met by the Swedish healthcare system.[83] To the best of our knowledge, this will be the first study to develop a Low-Intensity BA intervention for people with dementia in the Swedish setting. The proposed intervention has the potential to meet unmet psychological needs and also global priorities to support people with dementia to 'live well' with dementia and promote 'healthy ageing'. The intervention has further potential to reduce informal caregiver burden by providing effective strategies for informal caregivers to support people with dementia and depression.[31] The planned study is designed to improve the implementation potential of the intervention by utilising NPT. Given barriers to the implementation of evidence-based healthcare interventions are common, NPT will provide a framework for understanding potential barriers and facilitators to implementation from the intervention development phase. This will enhance future implementation potential should the intervention be demonstrated to be clinically and cost effective in the future. Our study design also allows us to collect in-depth qualitative data to provide a rich understanding of how to adapt the intervention from the perspective of people with dementia and informal caregivers. Our careful exploration of the perceived relevance of the intervention content and language and ways to enhance relevancy, cultural appropriateness, and acceptability, will facilitate the development of a more acceptable and relevant intervention, specifically tailored to needs and preferences of the population.

Despite the aforementioned study strengths, there are some limitations. First, people with dementia do not need to have past or present depression to participate. As such, we may recruit people with

dementia who have no experience of depression or low mood, thus limiting the transferability of findings to people with dementia and depression. However, we will ask people with dementia about their past and present wellbeing, providing us with some indication as to the experience of depression in the study sample. Second, due to resource limitations, we are unable to include ethnic minority groups who do not speak Swedish due to the increasing costs for e.g., recruitment, informed consent, and need for translated materials. This will limit the transferability of results and sample representativeness, e.g., for people with dementia and a migration background, who are at increased risk of marginalisation and unfortunately there is currently a lack of culturally appropriate interventions for this population.[84] However, the innovative intervention development process adopted in this study has potential for wider applications and can serve as a template for future adaptations of the intervention for people with dementia and other populations not represented in the present study.

In conclusion, this study will result in the development of a tailored intervention, hopefully optimised to improve relevancy and acceptability for a currently neglected population. Results will be used to inform the design of a phase II feasibility following the MRC framework for the development and evaluation of complex interventions[35] to further explore the feasibility and acceptability of the intervention.

Reviewer 2

Thank-you for your very helpful comments. Please find how we have addressed each comment below.

- 1 Good for reader to know, if the author can provide more information on the anticipated cultural barrier in adapting BA intervention from UK to Sweden. Any previous work(s) reported cultural barriers in adapting psychological intervention from UK to Sweden. Apart from language, what are the other cultural barriers. Any possible difference in knowledge, attitudes and practices (KAP) between people with dementia (also caregiver, other stakeholders) in UK and Sweden which may led to current proposed adaptation. This will be very interesting for readers, especially from outside Europe.**

We have added some additional information to the introduction regarding possible cultural differences between the English versus Swedish contexts (see page 6, lines 23-34 and page 7, lines 1-7):

Cultural adaptation of an intervention refers to the systematic modification of the intervention considering language, culture, and context to ensure its compatibility with the target population's cultural patterns, meanings, and values.[39] An example of a language differences is the meaning of the term 'depressed', which in Swedish is associated with a formal diagnosis of depression, whereas in English the term is used more colloquially.[40] Consideration of context can include a number of characteristics that could impact intervention delivery and effectiveness, and may include geographical, sociocultural, socioeconomic, ethical, legal, and political factors.[41-42] One contextual difference between England and Sweden is at present there is no psychological practitioner workforce in Sweden specifically trained in supporting LI-CBT.[43] As such, an appropriate healthcare professional workforce will need to be identified to facilitate intervention delivery. In addition, in England national non-profit organisations, such as Alzheimer's societies, provide support such as cognitive rehabilitation, information for informal caregivers, and activity groups, whereas in Sweden, support is predominantly provided via formal health and social care services rather than non-governmental organisations.[44] However, to the best of our knowledge, little research has examined potential cultural, language, and contextual differences between

England and Sweden in the context of psychological intervention development and adaptation. Consequently, it is difficult to anticipate in advance what cultural adaptations may be required.

2 *Not sure how researcher come out with required participant for their study. Healthcare professionals (n=8), community stakeholders (n=6), PwD (n=10), caregiver (n=10) etc. Any sample size calculation was carried out for focus group or rule of thumb approach? Please provide detail or justification how researchers determine those numbers.*

We have added a section "Sample size considerations" for clarification in the protocol manuscript (see page 9, lines 28-31 and page 10, lines 1-3).

Sample size considerations

Sample size will be guided by thematic data saturation and as such we cannot stipulate the exact sample size in advance.[60-61] Interviews and focus groups will be analysed iteratively and a decision concerning whether thematic saturation is met will be made during analysis. However, we anticipate to include n≈8 healthcare professionals, n≈8 community stakeholders in each focus group[62], and n≈10 people with dementia, and n≈10 informal caregivers respectively to participate in semi-structured interviews.

3 *Good to state the four settings at 'Setting' first.*

We have now stated the four settings as suggested (see page 8, lines 15-18):

People with dementia and informal caregivers will be recruited via four specialised health and social care service settings across the county of Uppsala: Primary Healthcare Centres; Memory Clinic; Day Care Centres; and via Dementia Care Consultants.

In addition, since the protocol has been submitted, we made an amendment to the protocol (and ethical application which has been approved) to recruit healthcare professionals and community stakeholders from across Sweden. Recruitment setting location was extended for pragmatic purposes, mainly around the recruitment of community stakeholders (see page 8, lines 20-21):

Healthcare professionals and community stakeholders from non-profit organisations will be recruited from locations across Sweden.

4 *Probable diagnosis? Not confirmed diagnosis? Please clarify because this protocol aims BA for depression in PwD.*

On reflection "probable diagnosis" was an unclear term to use. We will not confirm the diagnosis of dementia in participants' medical records. Instead, participants will self-report that they have a diagnosis of dementia. In order to maximize recruitment, we would like to identify people with dementia through a variety of health and social care settings. Adding an additional step of confirming the diagnosis (for example checking medical records) is a time and resource intensive procedure which may impact negatively on our ability to recruit participants in a timely manner. As such, we will rely on self-report data only. However, as we are only recruiting people with dementia (and their informal caregivers) from specialist dementia health and social care settings, we will minimise the risk of including people without dementia. In addition, participants recruited from Academic Healthcare Centres will have a diagnosis of dementia, as this will be screened by research nurses. We have removed the term "probable diagnosis" from the manuscript and replaced with "self-reported" (see page 9, line 1-2):

1. Have a self-reported diagnosis of dementia (any type). The research team will not confirm the diagnosis by checking medical records;

5 *Only person with mild to moderate dementia or this study will include those with severe dementia. It is challenging to implement intervention for those people with severe*

dementia. If only mild to moderate dementia, how the researcher determines the severity e.g., CDR? MMSE?

Previous research has shown that people with mild-to-moderate dementia most often have the capacity to consent. One of the inclusion criteria of our study is the person with dementia will need to have capacity to consent and therefore we anticipate recruiting of people with mild-to-moderate dementia. We have revised the manuscript accordingly (see page 9, lines 4-5 and page 18, lines 17-19):

Able to and have the capacity to provide informed consent, indicating mild to moderate dementia;[59]

Given only people with dementia with capacity to consent will be included in this study, we anticipate people with mild to moderate dementia will participate.

6 Any screening to identify the present of depression in PwD. How about level of depression? Although depression is common in PwD but not all PwD have depression. Any criteria? Because this proposal is BA for PwD and with Depression.

Thank you for raising this important consideration. For this first study (development and adaptation), we will not limit to only recruiting people with dementia and depression. For the planned future feasibility study and randomised controlled study, we aim to recruit people with dementia and depression. However, this is indeed a study limitation and might lead to the recruitment of people with dementia with no previous experience of depression/low mood. However, during the screening meeting, the person with dementia will complete a sociodemographic background form, including questions about well-being during the last 12 months as well as earlier in life, providing us with some information about their past and present experience of mental health difficulties. We have added this limitation in the protocol manuscript (see page 4, lines 11-13 and page 17, lines 23-28):

People with dementia do not need to have past or present depression to participate, hence we may recruit people with dementia who have no experience of depression or low mood, which may impact on transferability or results.

Despite the aforementioned study strengths, there are some limitations. First, people with dementia do not need to have past or present depression to participate. As such, we may recruit people with dementia who have no experience of depression or low mood, thus limiting the transferability of findings to people with dementia and depression. However, we will ask people with dementia about their past and present wellbeing, providing us with some indication as to the experience of depression in the study sample.

7 How about frailty level in PwD. Those people with dementia but robust & pre-frail may able to participate in BA program. Do physical condition of PwD matter for this protocol?

Frailty level in the person with dementia may represent a barrier for participation in a BA intervention. During the interviews with people with dementia and informal caregivers, we will ask about perceived barriers to intervention use and potential ways of overcoming these barriers. To improve clarity, we have now added some examples of potential barriers to the interview topic guide summary regarding barriers to intervention use for people with dementia and informal caregivers (see page 14, lines 2-3):

Barriers and facilitators to people with dementia and informal caregivers using the intervention (e.g., travel and mobility, physical health problems, stigma, informal caregiver resistance, COVID-19 restrictions);

8 Page 11, line 9 to 11 and line 22 to 24 are repetition on invitation/information pack. Please revise to avoid repetition.

We have removed the repetitive sentence (see page 10, line 24):

If permission is granted, a member of the research team will contact the person with dementia over the telephone to provide more study information and send a study invitation pack (as above) by post to interested potential participants.

9 Page 11, line 17-18; ' A research nurse....against the inclusion...' So, in medical record the research will screening suitable PwD for study? Should the researcher team screen against the inc/exc criteria after identifying study participant (PwD), right? Early in page 10 (Inc/exc criteria), stated e.g. 'self-reported misuse...' This only can establish after interview or talk with caregiver and PwD. Please improve the clarity.

This explanation was indeed too vague in the protocol. As you describe, a research nurse will conduct searches in medical records for suitable people with dementia against some of the inclusion criteria (the inclusion criteria visible in medical records), e.g., dementia diagnosis. When a person with dementia has shown interest, an eligibility screening meeting (preferably face-to-face) will be set up by the research team. During this meeting, people with dementia will complete an eligibility screening and background and sociodemographic questionnaire, including questions in relation to the 'remaining' inclusion/exclusion (those not covered in the searches conducted by the research nurse), e.g., self-reported misuse of alcohol etc. In case of self-reported misuse of alcohol etc., the person with dementia will be excluded from the study. In addition, during this meeting, the research team will assess the capacity of the person with dementia to consent. We have updated the protocol accordingly: (see page 10, lines 18-20, page 11, line 34 and page 12, line 1)

Primary healthcare centres who are part of the network of Academic Primary Healthcare Centres will appoint a research nurse to conduct patient record searches to identify potentially suitable people with dementia.

After provision of informed consent, an eligibility screening will take place with questions asked in accordance with the inclusion/exclusion criteria.

10 Page 13, line 7, please check the sentence's wording.

We have now re-worded this sentence (see page 12, lines 6-7):

Study invitation packs will include an anonymous reasons for non-participation form for those who decline participation.

11 Good if can provided a table with flow/data collection with timeframe/timing (days or weeks or months) for by Phase (1, 2, 3 & 4). Good to have or indicate study participants involved for each phase. Having this summary table will useful for reader.

We have added a figure describing the data collection and topics for the four PAR Phases, as well as adding estimated timeframe (see page 13, lines 11-13):

Four PAR Phases will be conducted, with estimated time frame of one month for each Phase. An overview of the data collection procedure and topics for the four PAR Phases can be found in Figure 2.

Figure 2. An overview of the data collection procedure and topics for the four PAR Phases.

12 What is s-NoMAD? Please give brief detail about that tool for reader e.g., items, scoring etc.?

We have added a brief explanation of the S-NoMAD according to your suggestion (see page 15, lines 9-15):

The S-NoMAD[74] consists of 20 items corresponding to the four NPT constructs (Coherence, Cognitive Participation, Collective Action, and Reflexive Monitoring)[47] in relation to the potential implementation of the proposed BA intervention. Responses are provided using a five-point Likert scale, ranging from “Strongly Disagree” to “Strongly Agree”. “Neutral” and “Not applicable” are also given as options.[74] The internal consistency of the S-NoMAD (Cronbach’s alpha 0.76 to 0.83) are in line with the original NoMAD.[75]

13 S-NoMAD is to triangulate qualitative findings? Not sure how, please provide brief information to reader.

We have added a brief explanation of the triangulation method according to your suggestions (see page 15, lines 31-34 and page 16, lines 1-2):

The S-NoMAD results will be analysed descriptively[74] in SPSS version 25. Triangulation by data source, where data sources are assessed against one another to cross-check data and aid interpretation,[77] will be conducted by exploring agreements and disagreements across the qualitative (interviews/focus groups) and quantitative (S-NoMAD) data. Comparisons between the qualitative and quantitative data sets will be discussed in the research team to explore potential reasons for discordant data.

Additional references

39. Bernal G, Jiménez-Chafey MI, Domenech Rodríguez MM. Cultural adaptation of treatments: a resource for considering culture in evidence-based practice. *Prof Psychol Res Pr* 2009;40:361-8. doi:10.1037/a0016401.
40. Grundström H, Rauden A, Olovsson M. Cross-cultural adaptation of the Swedish version of Endometriosis Health Profile-30. *J Obstet Gynaecol* 2020;40:969-73. doi:10.1080/01443615.2019.1676215.
41. Evans RE, Craig P, Hoddinott P, et al. When and how do 'effective' interventions need to be adapted and/or re-evaluated in new contexts? The need for guidance. *J Epidemiol Community Health* 2019;73:481-2. doi:10.1136/jech-2018-210840.
42. Pfadenhauer LM, Gerhardus A, Mozygemba K, et al. Making sense of complexity in context and implementation: the Context and Implementation of Complex Interventions (CICI) framework. *Implement Sci* 2017;12:21. doi:10.1186/s13012-017-0552-5.
44. Bieber A, Stephan A, Verbeek H, et al. Access to community care for people with dementia and their informal carers: case vignettes for a European comparison of structures and common pathways to formal care. *Z Gerontol Geriatr* 2018;51:530-6. doi:10.1007/s00391-017-1266-7.
57. Statistiska centralbyrån. Folkmängd i riket, län och kommuner 31 mars 2021 och befolkningsförändringar 1 januari - 31 mars 2021. Totalt. Stockholm: Statistiska centralbyrån; 2019. Available from: <https://www.scb.se/hitta-statistik/statistik-efter-amne/befolkning/befolkningens-sammansattning/befolkningsstatistik/pong/tabell-och-diagram/kvartals--och-halvarsstatistik--kommun-lan-och-riket/folkmangd-i-riket-lan-och-kommuner-31-mars-2021-och-befolkningsforandringar-1-januari---31-mars-2021.-totalt/>. [Accessed 2021 May 27].
59. Hegde S, Ellajosyula R. Capacity issues and decision-making in dementia. *Ann Indian Acad Neurol* 2016;19(Suppl 1):S34-S39. doi:10.4103/0972-2327.192890.
60. Saunders B, Sim J, Kingstone T, et al. Saturation in qualitative research: exploring its conceptualization and operationalization. *Qual Quant* 2018;52:1893-907. doi:10.1007/s11135-017-0574-8.
61. Sim J, Saunders B, Waterfield J, et al. Can sample size in qualitative research be determined a priori? *Int J Soc Res Methodol* 2018;21:619-34. doi:10.1080/13645579.2018.1454643.
62. Rabiee F. Focus-group interview and data analysis. *Proc Nutr Soc* 2004;63:655-60. doi:10.1079/pns2004399.

70. Farrand P, Woodford J, Anderson M, et al. Getting More Out of Every Day with Memory Difficulties: A Guide for People Living with Memory Difficulties. Exeter: University of Exeter 2015a.

71. Farrand P, Woodford J, Anderson M. Getting More Out of Every Day with Memory Difficulties: A Guide for Family and Friends. Exeter: University of Exeter 2015b.

72. Roth AD, Pilling S. Using an evidence-based methodology to identify the competences required to deliver effective cognitive and behavioural therapy for depression and anxiety disorders. *Behav Cogn Psychother* 2008;36:129-47. doi:10.1017/S1352465808004141.

75. Finch TL, Rapley T, Girling M, et al. Improving the normalization of complex interventions: measure development based on normalization process theory (NoMAD): study protocol. *Implement Sci* 2013;8:43. doi:10.1186/1748-5908-8-43.

84. Nielsen TR, Nielsen DS, Waldemar G. Barriers in access to dementia care in minority ethnic groups in Denmark: a qualitative study. *Aging Ment Health* 2020;1-9. doi:10.1080/13607863.2020.1787336.